# Merging or Computing Saturated Cost Partitionings?
# A Merge Strategy for the Merge-and-Shrink Framework

**Primary Keywords:** *None*

## Abstract

The merge-and-shrink framework is a powerful tool for computing abstraction heuristics for optimal classical planning. Merging is one of its name-giving transformations. It entails computing the product of two factors of a factored transition system. To decide which two factors to merge, the framework uses a merge strategy. While there exist many merge strategies, it is generally unclear what constitutes a strong merge strategy, and a previous analysis shows that there is still lots of room for improvement with existing merge strategies. In this paper, we devise a new scoring function for score-based merge strategies based on answering the question whether merging two factors has any benefits over computing saturated cost partitioning heuristics over the factors instead. Our experimental evaluation shows that our new merge strategy achieves state-of-the-art performance on IPC benchmarks.

## Introduction

*Classical planning* is the problem of finding a sequence of deterministic actions that lead from a given initial state to a state satisfying a desired goal condition (e.g., Ghallab, Nau, and Traverso 2004). The dominant approach of recent years to *optimally* solving classical planning problems is *heuristic search*, in particular using the A* algorithm in conjunction with *admissible heuristics* (Pearl 1984). The state-of-the-art class of admissible heuristics is based on *abstractions*, such as pattern databases (e.g., Rovner, Sievers, and Helmert 2019), domain abstractions (Kreft et al. 2023), Cartesian abstractions (e.g., Seipp and Helmert 2018), and *merge-and-shrink (M&S)* abstractions (e.g., Sievers and Helmert 2021), the latter being the focus of this work.

The M&S framework first computes a factored representation of the given planning task, called *factored transition system (FTS)*, which consists of transition systems (called *factors*) sharing the same set of labels. This FTS implicitly represents the state space of the task through its *product system*. The framework then repeatedly applies *transformations* to the current FTS. At any point, each factor of the FTS is an abstraction of the initial FTS (and with this, of the task).

One of the name-giving transformations is *merging*, which means to replace two factors by their product in the FTS. To decide which pair of factors to merge, the framework uses a *merge strategy*. Starting with the original work adapting merge-and-shrink from model check-

ing (Dräger, Finkbeiner, and Podelski 2009) to planning (Helmert, Haslum, and Hoffmann 2007; Helmert et al. 2014), there has been considerable work exploring merge strategies (Sievers, Wehrle, and Helmert 2014; Fan, Müller, and Holte 2014; Sievers et al. 2015; Sievers, Wehrle, and Helmert 2016). The current state of the art is constituted by the two *score-based* merge strategies DFP and sbMIASM, which choose the best pair of factors by computing scores for them, and by the SCC merge strategy, which first computes the strongly connected components (SCC) of the causal graph (Knoblock 1994) of the task to partition the state variables and then uses any score-based merge strategy for first computing a (product) transition system for each block before possibly further merging the resulting products.

In general, merging cannot decrease the heuristic quality of the abstraction represented by the current FTS. However, we observe that it may also not improve it compared to the information available when not merging the factors. In particular, when computing *saturated cost partitionings (SCPs)* (Seipp, Keller, and Helmert 2020) over the two factors in question leads to an equally-informed heuristic compared to merging, we can decide to avoid increasing the size of the FTS by merging them and potentially even stop the M&S computation early in favor of computing SCPs over the remaining factors. We devise a new score-based merge strategy based on this observation and experimentally show that it establishes a new state of the art on IPC benchmarks.

## Background

The M&S framework works on any *transition system* as long as it allows for a factored representation. Planning tasks in the SAS$^+$ formalism (Bäckström and Nebel 1995), which are defined over *finite-domain state variables* $V$, induce such transition systems $\mathcal{T} = \langle S, L, T, s_0, S_* \rangle$, where $S$ is the set of *states* (defined over $V$), $L$ is the set of *labels* $\ell$ with *cost* $cost(\ell) \in \mathbb{R}_0^+$, $S \times L \times S \subseteq T$ is the *transition relation*, $s_0$ is the initial state, and $S_* \subseteq S$ is the set of *goal states*. An $s$-plan for $\mathcal{T}$ is a path $\pi = \langle \ell_1, \ldots, \ell_n \rangle$ from state $s$ to some goal state from $S_*$. Its cost is $cost(\pi) = \sum_{i=1}^{n} cost(\ell_i)$. It is *optimal* if there is no $s$-plan with lower cost. A *plan* for $\mathcal{T}$ is an $s_0$-plan for $\mathcal{T}$. Optimal planning is the problem of finding an optimal plan or showing that no plan exists.

A *heuristic* $h_{\mathcal{T}} : S \mapsto \mathbb{R}_0^+$ for $\mathcal{T}$ maps a state $s \in S$ to an estimate of the cost of an $s$-plan for $\mathcal{T}$. By $h_{\mathcal{T}}^*$ we denote the

**Algorithm 1:** M&S algorithm extended to compute SCP heuristics and to stop early according to the merge strategy.

**Input:** FTS $F$
**Output:** Heuristic for $F$
1: **function** M&SWITHSCP($F$)
2: $\quad F' \leftarrow F, H \leftarrow \emptyset$
3: $\quad$ **while** not TERMINATE($F'$) **do**
4: $\quad\quad i, j \leftarrow$ MERGESTRATEGY($F'$)
5: $\quad\quad$ **if not** $i, j$ **then break**
6: $\quad\quad F' \leftarrow$ LABELREDUCTIONSTRATEGY($F'$)
7: $\quad\quad H \leftarrow H \cup h_\omega^{\text{SCP}}$
8: $\quad\quad F' \leftarrow$ SHRINKSTRATEGY($F', i, j$)
9: $\quad\quad F' \leftarrow (F' \setminus \{\mathcal{T}_i, \mathcal{T}_j\}) \cup \{\mathcal{T}_i \otimes \mathcal{T}_j\}$
10: $\quad\quad F' \leftarrow$ PRUNESTRATEGY($F', i \otimes j$)
11: $\quad$ **return** COMPUTEHEURISTIC($F', H$)

**Algorithm 2:** Score-based merge strategy.

**Input:** FTS $F$, merge candidates $M$, scoring functions $S$
**Output:** Merge candidate from $M$
1: **function** SCOREBASEDMERGESTRATEGY($F, M, S$)
2: $\quad$ **for** SCORINGFUNCTION $\in S$ **do**
3: $\quad\quad scores \leftarrow$ SCORINGFUNCTION($F, M$)
4: $\quad\quad M \leftarrow \arg\min_{m \in M} scores(m)$
5: $\quad\quad$ **if** $|M| = 1$ **then**
6: $\quad\quad\quad$ **return** single element from $M$

*perfect heuristic* for $\mathcal{T}$ which maps a state $s$ to the cost of an optimal $s$-plan for $\mathcal{T}$. $h_\mathcal{T}$ is *admissible* iff $h_\mathcal{T}(s) \leq h_\mathcal{T}^*(s)$ for all $s \in S$. We drop $\mathcal{T}$ if it is clear from context.

An *abstraction* for $\mathcal{T}$ is a function $\alpha : S \to S'$. It induces the *abstract transition system* $\mathcal{T}^\alpha = \langle S', L, \{\langle\alpha(s), \ell, \alpha(t)\rangle \mid \langle s, \ell, t\rangle \in T\}, \alpha(s_0), \{\alpha(s) \mid s \in S_*\}\rangle$. The *abstraction heuristic* for $\mathcal{T}$ induced by $\alpha$ is defined as $h_\mathcal{T}^\alpha = h_{\mathcal{T}^\alpha}^*$, i.e., as the perfect heuristic for the abstract transition system.

Given multiple admissible heuristics $H = \langle h_1, \ldots, h_n\rangle$ for $\mathcal{T}$, the cost functions $C = \langle cost_1, \ldots, cost_n\rangle$ form a *cost partition* if $\sum_{i=1}^n cost_i \leq cost$. We write $h(s, cost')$ for the evaluation of $h$ on $s$ using an alternative cost function $cost'$ instead of $cost$. The *cost-partitioned heuristic* $h_{H,C}(s) = \sum_{i=1}^n h_i(s, cost_i)$ is admissible (Katz and Domshlak 2010). *Saturated cost partitioning* (SCP) computes cost functions $C$ as follows, assuming any fixed order $\omega$ for the heuristics from $H$. It maintains a *remaining cost function rc* which is initialized to $rc_0 = cost$. In each iteration $i$ over the heuristics according to $\omega$, it computes $cost_i$ as the minimal cost function satisfying $h_i(s, rc_{i-1}) = h_i(s, cost_i)$ for all $s \in S$, called *saturated cost function*, which for abstraction heuristics is uniquely defined as $cost_i(\ell) = \max_{\langle s, \ell, t\rangle \in T}(h_i(s, rc_{i-1}) - h_i(t, rc_{i-1}))$ for all $\ell \in L$, and sets the remaining costs for the next iteration to $rc_i = rc_{i-1} - cost_i$. We write $h_\omega^{\text{SCP}}$ for the resulting SCP heuristic.

A *factored transition system (FTS)* $F = \langle \mathcal{T}^1, \ldots, \mathcal{T}^n\rangle$ consists of transition systems, called *factors*, sharing the same set of labels. Let $\mathcal{T}^i = \langle S^i, L, T^i, s_0^i, S_*^i\rangle$ for $1 \leq i \leq n$. $F$ compactly represents the *(synchronized) product* defined as $\bigotimes F = \langle S^\otimes, L, T^\otimes, s_0^\otimes, S_*^\otimes\rangle$, where $S^\otimes, s_0^\otimes, S_*^\otimes$ is the Cartesian product over the components of all factors $\mathcal{T}^i$ and $T^\otimes = \{\langle s^1, \ldots, s^n\rangle, \ell, \langle t^1, \ldots, t^n\rangle \mid \langle s^i, \ell, t^i\rangle \in T^i\}$.

Algorithm 1 shows the M&S framework as implemented in the Fast Downward planning system (Helmert 2006), extended with the facility to optionally compute SCP heuristics (Sievers et al. 2020). Ignore line 5 for the moment. For a given $F$, the algorithm runs its main loop until the maintained FTS $F'$ only contains a single factor or function TERMINATE stops the loop (line 3). In each iteration, it selects the pair of factors to merge next (line 4), possibly applies label reduction (line 6), which means abstracting the set of labels, possibly shrinks the two factors (line 8), which means abstracting them, merges the two factors (line 9), which means replacing the factors by their product in $F'$, and prunes the product (line 10), which means removing dead states and their transitions. All of these *transformations* apply abstractions to $F'$, and at any point, each factor $\mathcal{T}$ of $F'$ is an abstraction of the original FTS $F$ and as such induces the *factor heuristic* for $F$, written $h_F^\mathcal{T} = h_\mathcal{T}^*$.[1] At the end (line 11), the algorithm either returns the standard M&S heuristic $h^{\text{M&S}} = \max_{\mathcal{T} \in F'} h_F^\mathcal{T}$, defined as the maximum heuristic over the factor heuristics induced by $F'$, or the M&S-SCP heuristic $h_{\text{SCP}}^{\text{M&S}} = \max_{h \in H} h$, defined as the maximum heuristic over all SCP heuristics $h_\omega^{\text{SCP}} \in H$ previously computed (line 7) using some order $\omega$ over the factor heuristics induced by intermediate FTS $F'$.

A *merge strategy* needs to decide which pair of factors to merge given the FTS. We consider *score-based merge strategies* (Sievers, Wehrle, and Helmert 2016) that use *scoring functions* for evaluating merge candidates (i.e., pairs of factors) of an FTS. As shown in Algorithm 2, given an FTS $F$, a set of merge candidates $M$ over $F$, and some scoring functions $S$, the strategy iteratively (line 2) computes scores for all merge candidates using a scoring function (line 3), removes all but the best candidates (line 4), and repeats until only a single candidate is left which it returns (line 6). To ensure that a single merge candidate remains, at least one scoring function must define unique scores for distinct merge candidates. We also use the *SCC* merge strategy which initially partitions the variables of the task and during execution of the M&S algorithm uses score-based merge strategies to decide which factors within each block to merge next, before possibly also merging the resulting products afterwards.

## Merging or Computing Cost Partitions

Due to the large space of possible merge strategies, it is hard to find general criteria defining strong merge strategies, and the analysis by Sievers, Wehrle, and Helmert (2016) shows that state-of-the-art merge strategies still leave ample room for improvement. When using the M&S framework extended to compute the M&S-SCP heuristic, a natural question that arises is how merging two factors compares to leaving them for exploitation in the SCP(s) computed during M&S. To address this question, we devise the *maximum*

---

[1]Note that M&S uses special data structures to store the state mapping from the original FTS to individual factor heuristics (Helmert, Röger, and Sievers 2015). As the details do not matter, we omit them in the presentation.

| | Algorithm 3: Filter-based merge strategy. |
|---|---|

**Input:** FTS $F$, merge candidates $M$, filtering functions $S$
**Output:** Merge candidate from $M$ or None
1: **function** FILTERBASEDMERGESTRATEGY($F$, $M$, $S$)
2:     **for** FILTERINGFUNCTION $\in S$ **do**
3:         $M \leftarrow$ FILTERINGFUNCTION($F$, $M$)
4:         **if** $M = \emptyset$ **then**
5:             **return none**
6:     **return** single element from $M$

| | sf | | | | ff | | | |
|---|---|---|---|---|---|---|---|---|
| | mSCP | | mFactor | | mSCP | | mFactor | |
| | init | avg | init | avg | init | avg | init | avg |
| $h^{\text{M\&S}}$ | **902** | 875 | 889 | 859 | 793 | 857 | 871 | 836 |
| $h^{\text{M\&S}}_{\text{SCP}}$ | **990** | 909 | 916 | 901 | 953 | 908 | 907 | 917 |

Table 1: Coverage of the mFactor and mSCP scoring (sf) and filtering (ff) functions, using the initial (init) or the average (avg) heuristic value.

*SCP scoring function (mSCP-sf)* that prefers merge candidates whose product heuristic yields the largest improvement compared to the maximum over the two SCP heuristics over the two factors. Analogously, the *maximum factor scoring function (mFactor-sf)* prefers candidates whose product heuristic improves most compared to the maximum over the two factor heuristics, thus mimicking the computation of the standard M&S heuristic. To evaluate the improvement of heuristics, we compare the heuristic values of the initial state or the average values over the finite heuristic values, denoted by function AVG.

Formally, let $F = \langle \mathcal{T}^1, \ldots, \mathcal{T}^n \rangle$ be an FTS with $\mathcal{T}^i = \langle S^i, L, T^i, s_0^i, S_*^i \rangle$ for $1 \leq i \leq n$. Let $i, j \in \{1, \ldots, n\}$ with $i \neq j$, let $\mathcal{T}^{\otimes} = \mathcal{T}^i \otimes \mathcal{T}^j$, and let $s_0 = \langle s_0^1, \ldots, s_0^n \rangle$ be the initial state of $F$. Recall that $h_F^{\mathcal{T}^i}$, $h_F^{\mathcal{T}^j}$, and $h_F^{\mathcal{T}^{\otimes}}$ are the factor heuristics for $F$ induced by $\mathcal{T}^i$, $\mathcal{T}^j$, and $\mathcal{T}^{\otimes}$. We have the following variants for evaluating the merge candidate $\langle \mathcal{T}^i, \mathcal{T}^j \rangle$ and the product $\mathcal{T}^{\otimes}$:

$$h_{prod}^{init} = h_F^{\mathcal{T}^{\otimes}}(s_0)$$
$$h_{\text{mFactor}}^{init} = \max(h_F^{\mathcal{T}^i}(s_0), h_F^{\mathcal{T}^j}(s_0))$$
$$h_{\text{mSCP}}^{init} = \max(h_{\langle \mathcal{T}_F^i, \mathcal{T}_F^j \rangle}^{\text{SCP}}(s_0), h_{\langle \mathcal{T}_F^j, \mathcal{T}_F^i \rangle}^{\text{SCP}}(s_0))$$
$$h_{prod}^{avg} = \text{AVG}(h_F^{\mathcal{T}^{\otimes}})$$
$$h_{\text{mFactor}}^{avg} = \max(\text{AVG}(h_F^{\mathcal{T}^j}), \text{AVG}(h_F^{\mathcal{T}^i}))$$
$$h_{\text{mSCP}}^{avg} = \max(\text{AVG}(h_{\langle \mathcal{T}_F^i, \mathcal{T}_F^j \rangle}^{\text{SCP}}), \text{AVG}(h_{\langle \mathcal{T}_F^j, \mathcal{T}_F^i \rangle}^{\text{SCP}}))$$

Since we want to prefer candidates with the largest improvement of initial or average heuristic values of the product compared to the factors and since we need to minimize scores, we define mFactor-sf to compute the score as $h_{\text{mFactor}}^{init} - h_{prod}^{init}$ or $h_{\text{mFactor}}^{avg} - h_{prod}^{avg}$ depending on using initial or average heuristic values. Analogously, mSCP-sf is defined as $h_{\text{mSCP}}^{init} - h_{prod}^{init}$ or $h_{\text{mSCP}}^{avg} - h_{prod}^{avg}$.

In general, the difference computed by both scoring functions cannot be positive because merging, being an information-preserving transformation, dominates any other combination of the factor heuristics. However, in our implementation, we compute the product of the two *shrunk* factors to mimic what the M&S algorithm would do (cf. lines 8 and 9). This means that the difference computed by mSCP-sf can be positive, in which case merging is deemed worse than computing the SCP heuristics.

To accommodate situations in which for no pair of factors merging is deemed better than leaving them for exploitation in the maximum factor/SCP heuristic, we suggest a *filter-based* merge strategy. As shown in Algorithm 3, it iteratively (line 2) uses a *filtering function* to make the set of merge candidates smaller (line 3), returning none if all candidates have been filtered (line 5) or the single remaining candidate otherwise. Analogously to score-based merge strategies, at least one filtering function must uniquely determine a single candidate or discard all of them. We adapt the M&S algorithm to stop its computation when the merge strategy filtered all candidates, cf. line 5 of Algorithm 1.

Every score-based merge strategy can also be cast as a filter-based merge strategy by turning scoring functions into filtering functions that return the set of candidates with minimal score. Our maximum factor/SCP scoring functions, cast as filtering functions, additionally discard all merge candidates with a non-negative score. Furthermore, we extend the SCC merge strategy to allow using filter-based merge strategies instead of score-based ones and to return no merge candidate when the filtering functions discarded all candidates.

Finally, we remark that when stopping the M&S computation early due to mSCP-sf having discarded all candidates, $h_{\text{SCP}}^{\text{M\&S}}$ is not guaranteed to be at least as good as the heuristic we would obtain after continuing merging more factors. The reason is that SCP greedily assigns costs to factors so that not all pairs of factors can have assigned full costs in the SCP computed over the final FTS. We therefore also consider adding the SCP heuristics computed over all pairs of remaining factors to the set $H$ before computing $h_{\text{SCP}}^{\text{M\&S}}$.

## Experiments

We implemented all strategies in the existing M&S framework in Fast Downward 23.06 and evaluate them computing M&S and M&S-SCP heuristics for at most 900s, using bisimulation-based shrinking with a size limit of 50000 states, exact label reduction and full pruning of dead states. We evaluate the heuristics in an $A^*$ search, using Downward Lab (Seipp et al. 2017) to limit each planner run to 30 minutes and 3.5 GiB on IPC benchmarks from all sequential optimal tracks, a set consisting of 66 domains with 1847 tasks in total. Following Sievers et al. (2020), we compute an SCP heuristic in each iteration of the M&S algorithm using a random order over the factor heuristics.

We begin by evaluating mFactor and mSCP using initial (init) or average (avg) h-values, used as scoring (sf) or filtering (ff) functions in a score-based or filter-based merge strategy. Table 1 shows coverage, i.e., number of solved tasks,

| | mSCP-sf | | mSCP-ff | | |
| --- | --- | --- | --- | --- | --- |
| | none | alw | none | stop | alw |
| $h_{\text{SCP}}^{\text{M\&S}}$ | **990** | 982 | 953 | 948 | 948 |

Table 2: Coverage of the mSCP scoring (sf) and filtering (ff) functions using init, without (none) and with the addition of SCP heuristics computed over all pairs of remaining factors, either always (alw) or only if stopping M&S early (stop).

| | | | SCC | | mSCP-sf | | mSCP-ff | |
| --- | --- | --- | --- | --- | --- | --- | --- | --- |
| | DFP | sbM | DFP | sbM | | SCC | | SCC |
| $h^{\text{M\&S}}$ | 882 | 920 | 922 | 914 | 902 | **927** | 793 | 779 |
| $h_{\text{SCP}}^{\text{M\&S}}$ | 915 | 965 | 951 | 957 | 990 | **1006** | 953 | 943 |

Table 3: Coverage of state-of-the-art merge strategies and mSCP-sf/ff using init, including integration with SCC.

| | | | SCC | | mSCP-sf | | mSCP-ff | |
| --- | --- | --- | --- | --- | --- | --- | --- | --- |
| | DFP | sbM | DFP | sbM | | SCC | | SCC |
| DFP | – | 6 | 2 | 5 | 2 | 2 | 14 | 18 |
| sbM | **19** | – | **15** | 3 | 9 | 6 | 16 | **18** |
| SCC-DFP | **8** | 10 | – | 7 | 7 | 2 | 16 | **19** |
| SCC-sbM | **20** | **4** | **15** | – | 11 | 7 | 17 | **21** |
| mSCP-sf | **27** | **16** | **24** | **18** | – | 5 | **20** | 20 |
| +SCC | **27** | **19** | 21 | 17 | 7 | – | 19 | **22** |
| mSCP-ff | **26** | **17** | **22** | **18** | 5 | 7 | – | **4** |
| +SCC | **22** | 14 | **19** | 15 | 8 | 5 | 3 | – |

Table 4: Per-domain coverage of the same strategies as in Table 3, for $h_{\text{SCP}}^{\text{M\&S}}$ only. An entry in row $x$ and column $y$ denotes the number of domains in which $x$ solves more tasks than $y$. It is bold if $(x, y) \geq (y, x)$.

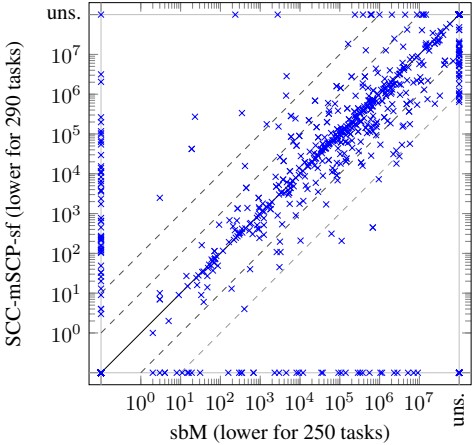

Figure 1: Expansions of sbM vs. SCC-mSCP-sf.

of all combinations. We observe that stopping the M&S algorithm when there is no good merge candidate (ff) leads to worse coverage, presumably because continuing merging factors can potentially lead to better factor heuristics in later iterations. We further observe that mSCP mostly dominates mFactor, likely because the evaluation of improvement is more nuanced with mSCP, except when terminating early and using $h^{\text{M\&S}}$, which seems reasonable given that $h^{\text{M\&S}}$ does not compute SCP heuristics. Finally, using the initial heuristic value to evaluate merge candidates is a better criterion than using the average heuristic value except for two cases of ff. In the remainder, we only consider the mSCP scoring and filtering functions using initial heuristic values.

Next, we evaluate the addition of SCP heuristics computed for each pair of remaining factors (one for each order) to the set $H$ before computing $h_{\text{SCP}}^{\text{M\&S}}$. For the filter-based strategy, we consider the alternatives of always adding these heuristics (alw) or only if the M&S algorithm stopped due to the merge strategy having filtered all candidates (stop). Table 2 shows coverage for these variants in comparison to not including these additional SCP heuristics (none). Clearly, there is no positive effect due to including the additional SCP heuristics. The likely reason is that SCPs over pairs of factors generally do not yield strong heuristics compared to SCPs over full FTS, so that the overhead caused by their inclusion is not worth it.

Finally, Table 3 shows coverage of the state-of-the art strategies DFP and sbMIASM (sbM), our best strategies with the maximum SCP scoring and filtering functions using initial heuristic values, and their integration with the SCC strategy. We observe again that the filter-based strategy cannot compete with the other strategies. While mSCP-sf solves fewer tasks than the state-of-the-art strategies when computing $h^{\text{M\&S}}$ (which seems reasonable given that $h^{\text{M\&S}}$ does not compute SCP heuristics), integrated with the SCC strategy, it outperforms them. For $h_{\text{SCP}}^{\text{M\&S}}$, both mSCP-sf and SCC-mSCP-sf significantly outperform the state of the art.

To verify that the strong coverage results do not stem only from a few domains, Table 4 compares the number of domains in which each planner in a row solves more tasks than the planners in the columns. We observe that both mSCP-sf and its integration with SCC strictly dominate all other strategies also under this measure. Finally, to assess where the strength of the new strategies stem from, Figure 1 compares the number of expansions of the $A^*$ search (excluding the last $f$-layer) using the previous best M&S-SCP heuristic computed with the sbM strategy to using our new best strategy. We note that while the heuristics display orthogonal strengths, there is a larger number of cases where our strategy results in a stronger heuristic than vice versa.

## Conclusions

We presented a scoring function for the M&S framework that prefers merge candidates whose product results in the largest heuristic improvement compared to using the factors in SCP heuristics instead. We also investigated filtering functions that stop the M&S algorithm if no merge candidate is deemed useful for merging. The new score-based merge strategy as well as its integration with the SCC merge strategy significantly outperform previous merge strategies. In future work, we want to investigate merge strategies which consider merging factors beyond a single iteration.

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
