# OpenReview forum: "Merging or Computing Saturated Cost Partitionings? A Merge Strategy for the Merge-and-Shrink Framework"
_icaps-conference.org/ICAPS/2024/Conference — ICAPS 2024_

### Official Review · Reviewer_W35a · 2024-01-20

**Significance And Importance:** 2
**Soundness:** 3
**Novelty:** 2
**Clarity:** 3
**Overall Evaluation:** 1
**Confidence:** 4

**Weaknesses:**

2: No major or minor weaknesses.

**Contributions Of The Paper:**

The paper introduces "greedy" scoring and filtering strategies for merging candidates in the M&S framework. Subsequently, it provides empirical evidence demonstrating that the scoring strategy outperforms the filtering-based approach.

**Ethical Considerations:**

(1) Not Applicable: The paper does not have any ethical considerations to address

**Nomination For Best Paper:**

No

**Questions For Authors:**

1. You write "using the initial heuristic value to evaluate merge candidates is a better criterion than using the average heuristic value except for two cases of ff."
Do you have an intuition about the reason for this?

2. I didn't see any mentioning about publishing the code upon the acceptance of the paper. I marked the option "Authors promise to release code and domains (whichever apply)."  Was I corrrect?

**Reproducibility:**

4: Authors promise to release code and domains (whichever apply).

**Strengths Of The Paper:**

The paper is well-organized and clearly written.
The  idea of the scoring function is novel (to my knowledge) and interesting.
The empirical results are compelling.

**Weaknesses Of The Paper:**

My only concern with this paper is the 'size' of its contribution, as it looks very close to the infamous least publishable unit.


Minor comments:

Should one cite the original work on A*? I don’t know, but given that there is enough space on the citations page, why not?

The Background section is technically sound. The question however, is how much information it provides for an unfamiliar reader? It seems that some details are missing in the synchronized product. In the text you use it as an unary operator on a set of transition systems, but in Alg 1 it is a binary operator on transition systems. Moreover, in line 10 of the algorithm you multiply what seems to be the indices i,j of transition systems. If this is a shorthand notation, please specify what is what.

You write “a set of merge candidates M over F”, does this means M is a subset of F, or does it mean that the elements in M were created using the factors of F, i.e., the elements of M can appear in F’ at some point?

Just to be sure, do you mean that h^{avg} := 1/|S| \sum_{s\in S} h(s), S are the states that can reach some goal state s_{\ast}? Can you make it clearer?

The results in table 4 are not easy to read. I understand that there are space limitations to present the large table, but domain-counting seems somewhat simplistic.


*POST-REBUTTAL*
===============
Thank you for your response.
I belive that the paper would benefit from a table of the per domain results. If purchasing an additional page is too expensive, consider publishing the results in supplementary materials and providing a citation to these materials in the original paper.
May the odds be ever in your favor.

---

> ### Author Rebuttal · Authors · 2024-01-27
>
> Thank you for your review!
>
> 1. We give some intuition for the general better suitability of the initial state
> estimates compared to average estimates as part of our response to question
> 4 of reviewer NfWb. We don't have a conclusive explanation for the two cases of
> ff. They both occur in a configuration where we combine a filtering function
> with the "wrong" M&S variant: mSCP with M&S (without SCP) and mFactor with
> M&S-SCP, so the filtering does not actually match the heuristic composition.
> Another observation is that early termination leads to weaker individual
> factors, so the scoring is based on comparisons of heuristics in a smaller
> range of estimates. With such small ranges the initial estimate and the average
> estimate tend to be very close and there is a larger variance which of the two
> is more accurate.
>
> 2. We will publish the code, experiment scripts, benchmarks and data.
>
> other:
>
> Regarding the size of the contribution, we agree that it would not necessarily
> justify a *long* paper. We still think that the aspects that we explore are
> important ones in the overall configuration space of the merge and shrink
> framework, in particular in the light of the recent integration of saturated
> cost partitioning. We also would like to point out that we significantly
> improve the state of the art in M&S, even improving in the well-explored
> standard M&S setting (without cost partitioning).
>
> We will include the reference for $A^*$.
>
> We abused notation for the synchronized product by applying it to (a
> set of) two factors and will clarify this notation. We will also fix line 10,
> as indicated in the rebuttal to reviewer NfWb.
>
> A merge candidate is comprised by the two factors that should potentially be
> merged, thus the set of merge candidates M contains all subsets of set F of
> cardinality 2.
>
> Your interpretation of $h^\mathrm{avg}$ is correct. Note that S consists of the abstract
> states that are reachable from the abstract initial state and from which in
> addition we can reach an abstract goal state.

---

### Official Review · Reviewer_NfWB · 2024-01-22

**Significance And Importance:** 2
**Soundness:** 3
**Novelty:** 2
**Clarity:** 3
**Overall Evaluation:** 1
**Confidence:** 4

**Weaknesses:**

1: Minor weaknesses that are easily fixable.

**Contributions Of The Paper:**

The paper proposes a new merge strategy for the well known frame and shrink framework.

The main contribution of the paper is a new scoring function to decide which two factors to merge next. This new scoring function prioritizes merging factors whose synchronized product produce the largest improvements compared to doing SCP with the two possible orders.

The paper also suggests a "filter-based" merge strategy, which instead of computing a scoring function for all merge candidates, filters candidates, allowing an early termination of the M&S loop if no merge candidate is deemed worthy of merging. This means that if given infinite resources, the M&S algorithm might finish with more than one factor.

The strategies were implemented in the existing M&S framework and evaluated in the IPC benchmarks from the sequential optimal track.

**Ethical Considerations:**

(5) Excellent: The paper comprehensively addresses all of the applicable ethical considerations

**Nomination For Best Paper:**

No

**Questions For Authors:**

1. Can you clarify the paragraph starting with "Finally, we remark that when stopping the M&S computation ... (lines 217-225)?
2. Can you solve the issues related to Alg.1 described in the "weakenesses of the paper"?
3. Can you clarify why if the order is relevant for scoring functions they are represented as a set in Alg. 2?
4. Can you clarify the details on the experiments: why do you have to presume the causes of the performance for the filtering-based approach? can you describe all the scoring functions used in the experiments (configuration)? Can you explain why using the average heuristic value is worse than the initial heuristic value? Can you include per domain results in coverage?
5. Do you plan to make the code publicly available?
5. Are you using the benchmarks from the last IPC or just those until IPC 2018?

**Reproducibility:**

2: Some details are missing, but the paper still appears to be replicable with some effort.

**Strengths Of The Paper:**

- The paper proposes an interesting new scoring function for the M&S framework.
- The paper is well-written.
- The idea of deciding automatically whether merging two factors has benefits over computing saturated cost partitioning over the factors makes many sense for M&S under SCP.
- The results are quite good.

**Weaknesses Of The Paper:**

The paper has some clarity issues.

Algorithm 1 has a couple of issues:
- If the merge strategy does not produce any i,j (first iteration, lines 4 and 5) H will be returned as the empty set (line 11). Shouldn't H be initialized as H=\{h^{SCP}_w\}?

- Line 7 uses a w order that's not specified in the algorithm. From what I understand you're always using random orders in your experiments,
but there's no reason why you couldn't use a different strategy. Shouldn't there be an order computation for the scp step?

- On line 10, you compute the synchronized product of two indices i,j instead of transition systems T_i, T_j. This is confusing.

Line 117 has confusing notation.

In Algorithm 2, the scoring functions seem to be represented as a set, but it should be made clear that there's an order to them. Different
orders can produce drastically different results.

Instead of using the argmin and making sure you have a scoring functions that defines unique scores, couldn't you use some kind of
tiebreaking considering all scoring functions? Or a voting system of some sort?

Merging or Computing Cost Partitions:
Lines 175-180: I assume you refer to the average values over all states, but maybe this could be explicitly written.

On Algorithm 3 the order of the filtering functions is relevant too, but they're being treated as a set.

When filtering, you discard all candidates that have a non-negative score. This seems to be quite aggressive, and maybe short-sighted:    maybe in this iteration merging two candidates is not beneficial, but subsequent iterations might greatly improve the end result. The motivation for this aggressive behavior might be lacking. For this filtering based merge strategy to be well justified for me, it should be less greedy, and devise ways of estimating if no further improvements can be made (farther than in the current iteration).

 What do you mean when you say "we extend the SCC merge strategy to allow using filter-based merge strategies instead of score-based ones and to return no merge candidate when the filtering functions discarded all candidates."? You compute the SCC and only consider candidates in the same component for the filtering functions? Isn't this again too aggressive?

Lines 217 through 225 are not clear to me. What do you mean when you say mSCP-sf discarded all candidates? Isn't sf refering to scoring    function? From what I understand the score-based merge strategy never discards all merge candidates. I guess this was suppossed to be    for mSCP-ff.  Assuming this refers to filtering-based merge strategies, I still do not understand what you mean by "We therefore also consider adding the SCP heuristics computed over all pairs of remaining factors to the set H before computing h^{M&D}_{SCP}".  If I'm missing something and this refers to score-based merge strategies, I think this should be explained clearer and in more detail, since  they produce the best results by a big margin.

Experiments:
The experiments show that the filtering approach yields worst results, and the explanation is "presumably because continuing merging     factors can potentially lead to better factors heuristics in later iterations". Why do you have to presume it? Can't you look at the same configurations for the score based filtering and verify this fact? The motivation and analysis of the results for the filtering based merge strategies are vague.

 What other scoring/filtering functions are you using? You only refer to one scoring function per configuration, but you need at least one of the scoring functions used in each configuration to produce unique scores, and none of the proposed scoring functions is  guaranteed to produce unique scores. I think more details about the configuration of each strategy would benefit the paper.

Intuitively, using the average h values should be better than using the initial heuristic value. Do you have an explanation of why using the initial h values is better?

The results are quite good. Can't you include a per domain results of the coverage? (Table 4 caption says per-domain coverage, but this    table is aggregating all domains).

Overall, the paper proposes an interesting new scoring functions for the M&S framework, but the motivation for filter-based merge strategy might be too vague. There are a few sections that could be explained in more detail. This might be because 2 out of 4 pages are for  introduction and background, leaving you with little space and having to be too brief in some sections.

-------------------
Post-rebuttal
-------------------
Thank you for your answers!
If accepted I would recommend to: (1) revise Algorithm 1 carefully. What is explained is that for the SCP heuristic it generates the max of the SCP heuristics in H, but it also do other things that are only explained in the text and that it would be good to define in a more formal way (scp over the final factors, including all possible pairs after ending the main loop); (2) incorporate per domain results (supplementary material or extra page).

---

> ### Author Rebuttal · Authors · 2024-01-27
>
> Thank you for your review!
>
> 1. Thanks for noting that this refers to mSCP-ff. It bases its decisions on
> comparing the product heuristic to the maximum over the two SCP heuristics
> computed for the two orders of the same two factors. But we don't include these
> two SCP heuristics in the set H of heuristics computed by M&S-SCP. Instead, we
> add a single SCP heuristic computed over *all* factors of the current FTSs to
> H in each iteration. This discrepancy is particularly critical if filtering
> leads to an early termination of M&S because H does not include the heuristic
> that motivated termination. We mitigate this by including all pairs after
> ending the main loop of M&S.
>
> 2. Thanks for pointing out these issues, we will fix them.
>
> The algorithm does not include the SCP over all initial factors by default, but
> computes an SCP over the remaining factors after stopping the main loop.
>
> In the experiments, w is always random (the SOTA as reported by Sievers et al.
> IJCAI 2020). Using different strategies is orthogonal to our work.
>
> We will clarify l10. Pruning may consider what factor was added last (and
> typically prunes that one).
>
> 3. The order is relevant, we will adapt the pseudo-code (dito Alg. 3)
>
> 4. You are right, the corresponding scoring and filtering-based strategies are
> identical except for early termination and indeed the results show that our
> presumption actually must be the reason.
>
> sbMIASM computes for two factors the ratio of the size of the product after and
> before pruning and prefers merges that permits most pruning. While the details
> of DFP are involved, the idea is to prefer merging factors that must
> synchronize on labels that occur close to a goal state (with the intention of
> a more fine-grained abstraction in the goal region). For tie-breaking, we use
> a scoring function that puts the factors in a total order and prefers the
> lexicographically smallest pair (following Sievers et al. ICAPS 2016) in all
> experiments.
>
> The heuristic value of the initial state is a somewhat accurate measure of
> quality for the search because it will definitively be expanded and the
> estimates on the reachable states cannot arbitrarily decrease (having
> a consistent heuristic). In contrast, many abstract states are never
> encountered during search, but they impact the average.
>
> If the reviewers find it helpful and we can buy an additional page, we can
> include per domain results (cannot include it here for space reasons).
>
> 5. yes
>
> 6. IPC'23 benchmarks are included.

---

### Official Review · Reviewer_mHhZ · 2024-01-22

**Significance And Importance:** 1
**Soundness:** 3
**Novelty:** 1
**Clarity:** 4
**Overall Evaluation:** 1
**Confidence:** 4

**Weaknesses:**

0: Minor weaknesses requiring some work to be addressed for the paper to be accepted.

**Contributions Of The Paper:**

The paper proposes a new scoring function for the M&S framework that uses the scoring-based merge strategy. It evaluates whether combining two factors improves the heuristic quality over keeping them separate for the saturated cost partitioning (SCP) heuristic.

The paper proposes two scoring functions, maximum SCP scoring function(mSCP-sf) and maximum Factor scoring function (mFactor-sf), which prefer merge candidates(pair of factors) whose product improves the heuristic the most in comparison to SCP and Factor heuristics, respectively, over the two factors.

The resulting heuristics are shown to have better coverage results than the previous state-of-the-art merge strategies DFP and sbM.

The paper explores how the mSCP and mFactor scoring functions perform under different settings, such as combining them with the SCC merge strategy, which partitions the state variables based on the strongly connected components of the causal graph of the task.

**Ethical Considerations:**

(5) Excellent: The paper comprehensively addresses all of the applicable ethical considerations

**Nomination For Best Paper:**

No

**Questions For Authors:**

I request you to respond to the concern raised in the weakness section.

**Reproducibility:**

3: Authors describe the implementation and domains in sufficient detail.

**Strengths Of The Paper:**

The proposed scoring functions for the merge strategy are shown to have better empirical performance than the previous state-of-the-art merge strategies.

The paper is well-structured. The contributions are well-explained and straightforward to understand.

**Weaknesses Of The Paper:**

The primary concern I have is that the paper does not justify the proposed scoring strategy well. The paper only provides empirical results to support its claim but does not explain the underlying logic or rationale behind the strategy. It is unclear why this strategy is superior to other alternatives. A more convincing justification would require a discussion of the implications and trade-offs of the strategy and theoretical considerations.

**POST-REBUTTAL
===============
Thank you for your responses.

---

> ### Author Rebuttal · Authors · 2024-01-27
>
> Thank you for your review!
>
> The theory of M&S shows that merging can never decrease the quality of the
> resulting standard M&S heuristic, i.e., the heuristic induced by the product of
> the two factors dominates the heuristics induced by each of the two factors (=
> factor heuristics) and thus also their maximum. However, the dominance is not
> necessarily strict (actually improving the heuristic). Connecting the theory to
> the M&S algorithm, we note that factors get potentially shrunk before merging
> and the product of the shrunk factors (= product heuristic) can be dominated by
> an individual factor. In such cases, it might be worth not to merge these two
> factors but to save resources for other merge transformations.
>
> The scoring function mFactor compares the product heuristic to the maximum of
> the two factor heuristics (= maximum factor heuristic) in terms of either the
> initial state heuristic value or the average heuristic value over all abstract
> states. The score is the difference between the maximum factor and the product
> heuristics and the merge strategy prefers to merge two factors which improve
> the M&S heuristic induced by the current FTS the most. So the rationale behind
> mFactor is a greedy decision for the best immediate improvement without looking
> ahead to the future transformations by M&S. Other scoring functions use other
> criteria, not directly targeting the immediate improvement. This possibly
> explains why mFactor scoring is better (in practice) than other scoring
> functions.
>
> If we apply the rationale behind mFactor-sf in M&S-SCP, we need to consider
> that the latter does not use the maximum factor heuristic but instead computes
> a SCP of the factor heuristics. Therefore, mSCP-sf also compares the product
> heuristic to the best of the two possible SCPs over the two factor heuristics.
> So it is just an adaptation of the same idea to the integration of cost
> partitioning into M&S.
>
> Score-based merging cannot stop M&S if the greedy decision does not actually
> improve the heuristic estimates, which motivates the filter-based strategy. Our
> two scoring functions can directly be cast as filtering functions, discarding
> all products of two factors that decrease heuristic quality rather than
> increasing it. The experiments show that this "aggressive" behavior (reviewer
> NfWb) indeed is often too short-sighted.
>
> If the reviewers agree that these explanations are helpful (and we can buy an
> additional page), we would gladly integrate them in the camera-ready copy.

---

### Meta-Review · Area_Chair_BSLx · 2024-02-04

**Recommendation:** Accept (Poster)
**Confidence:** 4

**Metareview:**

The authors present a new merging strategy for the merge-and-shrink framework, which is based on saturated cost partitioning. The idea is novel and shows promising results. It is not the most surprising result and is generally incremental work based on the current state of the art and already known ideas.
We feel that the result and this particular combination of MS and SCP is nevertheless worth publishing.

One main point I want to stress to the authors (if the paper is ultimately accepted) is readability. Especially details in algorithm 1 and the exact handling of the scps (and factor orders omega) should be better explained in the paper. For example, the author's statement "we mitigate this by including all pairs after ending the main loop of M&S" in their rebuttal should be explained in the paper in more detail.

**Ethical Considerations:**

(1) Not Applicable: The paper does not have any ethical considerations to address